 **eLIFE**

# Phenotypic outcomes in Mouse and Human *Foxc1* dependent Dandy-Walker cerebellar malformation suggest shared mechanisms

Parthiv Haldipur[1], Derek Dang[1], Kimberly A Aldinger[1], Olivia K Janson[1], Fabien Guimiot[2], Homa Adle-Biasette[2], William B Dobyns[1,3], Joseph R Siebert[4,5], Rosa Russo[6], Kathleen J Millen[1,3]*

[1]Center for Integrative Brain Research, Seattle Children's Research Institute, Seattle, United States; [2]Hôpital Robert-Debré, INSERM UMR 1141, Paris, France; [3]Department of Pediatrics, Genetics Division, University of Washington, Seattle, United States; [4]Department of Laboratories, Seattle Children's Hospital, Seattle, United States; [5]Department of Pathology, University of Washington, Seattle, United States; [6]Department of Pathology, Molecular Genetics Laboratory, University Medical Hospital, Salerno, Italy

**Abstract** *FOXC1* loss contributes to Dandy-Walker malformation (DWM), a common human cerebellar malformation. Previously, we found that complete *Foxc1* loss leads to aberrations in proliferation, neuronal differentiation and migration in the embryonic mouse cerebellum (*Haldipur et al., 2014*). We now demonstrate that hypomorphic *Foxc1* mutant mice have granule and Purkinje cell abnormalities causing subsequent disruptions in postnatal cerebellar foliation and lamination. Particularly striking is the presence of a partially formed posterior lobule which echoes the posterior vermis DW 'tail sign' observed in human imaging studies. Lineage tracing experiments in *Foxc1* mutant mouse cerebella indicate that aberrant migration of granule cell progenitors destined to form the posterior-most lobule causes this unique phenotype. Analyses of rare human del chr 6p25 fetal cerebella demonstrate extensive phenotypic overlap with our *Foxc1* mutant mouse models, validating our DWM models and demonstrating that many key mechanisms controlling cerebellar development are likely conserved between mouse and human.

*For correspondence: kathleen.millen@seattlechildrens.org

**Competing interests:** The authors declare that no competing interests exist.

## Introduction

The developmental pathology of Dandy-Walker Malformation (DWM), a common human cerebellar birth defect has not been completely delineated and only a few genetic causes have been identified (*Aldinger et al., 2009*; *Blank et al., 2011*; *Doherty et al., 2013*; *Aldinger and Doherty, 2016*). We previously reported that rare heterozygous deletions (del) of the forkhead box C1 (*FOXC1*) gene on human chromosome (chr) 6p25 are associated with DWM (2). We have also conducted an extensive phenotypic analysis of the *Foxc1* homozygous null (*Foxc1[-/-]*) mice where we have shown that both zones of neurogenesis, the ventricular zone and rhombic lip, are disrupted in *Foxc1[-/-]* mice (*Aldinger et al., 2009*; *Haldipur et al., 2014*). This analysis provided some key insights regarding the developmental disruptions underlying this important brain malformation, yet many questions remain.

Here, we present a developmental analysis of the foliation defects in *Foxc1* hypomorphic mutant (*Foxc1[hith/hith]*) mice. These mice are viable as adults and display a unique posterior foliation defect

that is strikingly similar to the DWM 'tail sign' which has been recently proposed to represent a pathognomonic feature of DWM (*Bernardo et al., 2015*). We present mouse genetic lineage mapping data that identifies premature and abnormal migration of posterior vermis-fated rhombic lip (RL) derived cells as the cause for this striking phenotype. We also present the analyses of 3 rare human fetal del chr 6p25 DWM cases which show extensive phenotypic overlap with *Foxc1* mutant mouse cerebellar defects, including ectopic granule cell progenitors (GCPs) and Purkinje cells, dysmorphic Bergmann glial fibers, and posterior vermis disorganization.

Since *Foxc1* mutations in mice recapitulate multiple aspects of the developing and mature human del chr 6p25 DWM cerebellar pathology, these data validate *Foxc1* mutant mice as models for human DWM. Further, our analyses clearly demonstrate that competent meningeal signaling is required for multiple aspects of prenatal and postnatal cerebellar development.

## Results

### Postnatal human del chr 6p25 (*FOXC1 +/-*) Dandy-Walker patients and *Foxc1*$^{hith/hith}$ mice share similar deficits in posterior vermis foliation

One hallmark of human DWM is posterior vermis hypoplasia. As illustrated in *Figure 1A–B*, the normal human cerebellar vermis has a trilobar appearance, with primary and secondary fissures readily distinguishable (*Figure 1B*, white arrowheads). The posterior vermis lobules, below the secondary fissure, are well formed and a distinct choroid plexus is evident (*Figure 1B*, yellow arrowhead). In contrast, the cerebellar vermis in our del chr 6p25 DWM patient cohort (*Aldinger et al., 2009*) ranges from very hypoplastic and dysplastic (*Figure 1C,D*), to less severely affected (*Figure 1E,F*). Regardless of the extent of dysplasia, these individuals with del chr 6p25 share a common extended and dysplastic posterior vermis with an indistinct choroid plexus. This feature has recently been designated the DWM 'tail sign' and may be pathognomonic for DWM (*Figure 1D,F*, red arrowhead) (*Bernardo et al., 2015*).

*Foxc1*$^{-/-}$ mice have substantial cerebellar abnormalities as neonates (*Aldinger et al., 2009*; *Haldipur et al., 2014*). During embryogenesis, *Foxc1*$^{-/-}$ mice have dramatic VZ proliferative deficits and GABAergic neuronal migration defects in addition to RL abnormalities (*Aldinger et al., 2009*; *Haldipur et al., 2014*). Neonatal lethality precludes evaluation of cerebellar foliation, which is a postnatal process in mice. Furthermore, mouse *Foxc1*$^{-/-}$ phenotypes are more severe than those seen in del chr 6p25 patients who retain one functional *FOXC1* allele. We previously reported that mice homozygous for a *Foxc1*$^{hith/hith}$ hypomorphic allele are viable as adults and have abnormal cerebellar foliation (*Aldinger et al., 2009*). These mice retain 5% Foxc1 activity (*Zarbalis et al., 2007*). Here, we add to our previous report, showing that *Foxc1*$^{hith/hith}$ mice have a highly unusual, partially formed posterior lobule (lobule X), which aberrantly exposes the internal granule layer to the fourth ventricle. This phenotype is reminiscent of the human DWM tail (*Figure 1I,J*). To our knowledge, *Foxc1*$^{hith/hith}$ mice are the only mouse model to consistently show this foliation phenotype. Thus, an analysis of the developmental events resulting in this highly specific and disease-relevant foliation defect are likely to inform the developmental pathology of the posterior vermis defects in human del chr 6p25 DWM.

### *Foxc1*$^{hith/hith}$ posterior defects are primarily caused by aberrant migration of the posterior-fated rhombic lip descendants

Cerebellar foliation normally follows a stereotypic pattern and sequence. Four cardinal vermis fissures, including the primary and secondary fissures, are anchored late in embryonic mouse development, then additional fissure formation and folia lengthening progress postnatally (*Sudarov and Joyner, 2007*; *Legué et al., 2015*). Although the primary and secondary fissures were readily evident at P15 (Postnatal day 15) in both wild-type and *Foxc1*$^{hith/hith}$ mice, a delay in posterior fissure formation was evident at P1 in *Foxc1*$^{hith/hith}$ mice, (*Figure 1M,N*; asterisks) and the *Foxc1*$^{hith/hith}$ posterior vermis was clearly disorganized at this stage (*Figure 1N*). At e17.5, the *Foxc1*$^{hith/hith}$ RL was very disorganized with an abnormal abundance of apparent GCP (*Figure 1P*) (Ki67+/Pax6+ data not shown), similar to the RL phenotype we previously reported in embryonic *Foxc1*$^{-/-}$ mice (*Aldinger et al., 2009*; *Haldipur et al., 2014*).

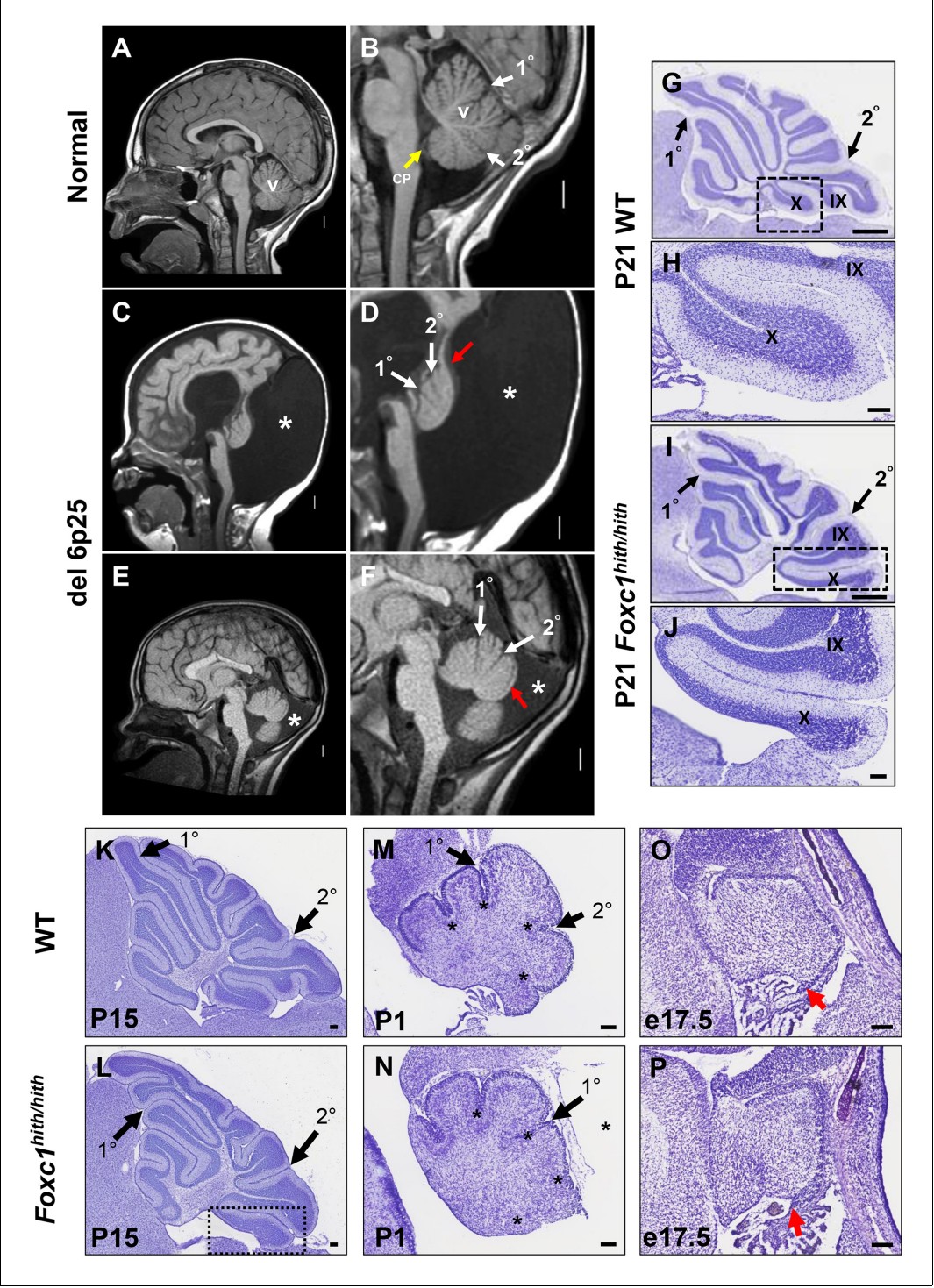

**Figure 1.** Similarities of human DWM and mouse *Foxc1*<sup>hith/hith</sup> posterior folial abnormalities. (A–F) T1-weighted midsagittal magnetic resonance images in the postnatal control subject (**A,B**) and two subjects with del chr 6p25.3 CNVs that include *FOXC1* or intragenic mutations of *FOXC1* diagnosed with Dandy Walker Malformation (*Aldinger et al., 2009*) (**C–F**). The midline cerebellar vermis (v) and choroid plexus (CP) are marked only in the controls. Asterisks (*) indicate an enlarged posterior fossa in DW cases. The white arrowheads mark the $1^0$ and $2^0$ fissures, while the red arrowhead indicates upward rotation of the cerebellar vermis and abnormal posterior DW tail. Sagittal sections of P21 cerebellar vermis from wild-type (**G,H, K, M, O**) and *Foxc1*<sup>hith/hith</sup> (**I,J, L, N, P**) mice. The *Foxc1*<sup>hith/hith</sup> cerebellum is characterized by the presence of a partially formed posterior lobule X (**I,L** box; **J**). The stereotypical wild-type cerebellum foliation pattern is disrupted in *Foxc1*<sup>hith/hith</sup> mutants. Primary and

*Figure 1 continued on next page*

*Figure 1 continued*

secondary fissures are noted (black arrowheads). Four cardinal fissures (black asterisks) divide wild-type postnatal cerebellar vermis into five cardinal lobes. Foxc1*hith/hith* mice exhibit an excess of granule cell progenitors (GCPs) in the e17.5 RL (P, red arrowhead). *Scale bars = 100 μm (H, J, K–P) and 500 μm (G,I).*

To further dissect the mouse *Foxc1^hith/hith* RL and posterior vermis phenotype, we employed a RL genetic fate mapping system. Specifically, we generated wild-type, *Foxc1^-/-*, and *Foxc1^hith/hith* mice carrying both an *Lmx1a-cre* transgene and *Ai14*, a tdTomato-cre reporter construct (*Chizhikov et al., 2010*; *Madisen et al., 2010*). In e17.5 WT animals (*Figure 2A*), *Lmx1a-cre* is expressed in the subset of embryonic cerebellar RL cells fated to give rise to the fourth ventricle choroid plexus epithelium, GCPs, and their descendants which are restricted to the posterior vermis (lobule X and the posterior half of lobule IX), and unipolar brush cells within the core of the developing cerebellum (*Chizhikov et al., 2010*). As expected, by P0, the RL had fully regressed in wild-type animals and the RL-derived *Lmx1a*-labeled lineages were clearly limited to lobule X and the posterior half of lobule IX (*Figure 2C*).

We next determined that Lmx1a expression was not dependent on Foxc1 expression. At the onset of Lmx1a RL expression at e13.5, there was no difference in Lmx1a expression between WT and *Foxc1^-/-*(*Figure 2—figure supplement 1A*). By e17.5, however, in *Foxc1^hith/hith* mice, *Lmx1a-cre* marked cells remained restricted to the posterior vermis yet the RL retained an excess of labeled cells (*Figure 2B*). Many streams of *Lmx1a*-lineage marked cells were also aberrantly located within the core of the developing cerebellum in addition to cells normally traversing into the External Granule Layer (EGL) (*Figure 2B*, arrows). By P0, labelled EGL cells had ectopically migrated anteriorly to the secondary fissure, the anterior boundary of lobule IX in the *Foxc1^hith/hith* mutants (*Figure 2D*). We also observed large numbers of *Lmx1a*-fate mapped cells along a residual posterior ventricular zone in *Foxc1^hith/hith* mutants (*Figure 2D*, yellow arrow). To determine if similar migration deficits contributed to the severe RL and early EGL phenotypes we previously reported in *Foxc1^-/-* mutant embryos, we also conducted *Lmx1a-cre* fate mapping on *Foxc1^-/-* mice (*Figure 2E–J*). As expected, in wild-type embryos, most *Lmx1a-cre* labelled cells exited the RL and migrated into the posterior EGL between e14.5 and e17.5 (*Figure 2E,G,I*). In *Foxc1^-/-*mutants, however, dramatic mismigration of the posterior-fated RL lineage was evident as early as e14.5 (*Figure 2F*). By e17.5, an abnormally large RL showed numerous retained *Lmx1a-cre+* cells. Ectopic *Lmx1a-cre+* streams of cells were also present along the VZ (*Figure 2H,J*). Quantification of tdTomato+ cells present outside of the EGL and RL area in the wild-type and *Foxc1^-/-*mutants indicates significantly higher numbers of ectopic tdTomato+ cells in the mutant (*Figure 2—figure supplement 1B*). Thus, RL and VZ development in *Foxc1^-/-* mutants is more severely disrupted than in *Foxc1^hith/hith* (*Figure 2—figure supplement 1C*) (*Haldipur et al., 2014*).

We tested if *Lmx1a*-fate mapped cells found in the e17.5 *Foxc1^-/-*cerebellar anlage core were incorrectly derived from re-specified VZ fates. We did not however detect any *Lmx1a*-lineage marked (tdTomato+) VZ progenitors (Sox9+), GABAergic interneurons (Pax2+) or Purkinje cells (Skor2+) (*Figure 2K–M*). Rather, all tdTomato+ cells retained a Pax6+ RL lineage identity (*Figure 2O–P*, arrows). As expected, a subset of the internally-located tdTomato+ *Lmx1a*-fate mapped cells was Tbr2+ positive (*Figure 2N,n*, arrows), consistent with the unipolar brush cell fate expected from the RL Lmx1a+ lineage.

However, we note that while a large subset of the misrouted GCPs continues to proliferate (*Figure 2Q*, red and yellow arrow; *Figure 2—figure supplement 2C,D*, arrows), we also observe that many of the *Lmx1a-cre* tdTomato+ cells that fail to migrate out of the RL eventually differentiate precociously within the RL (*Figure 2R*). We conclude that a combination of mismigration and precocious differentiation causes a depletion of granule cell progenitors required to form the posterior vermis, thus leading to severely hypoplastic posterior lobules in *Foxc1^hith/hith* mice.

The late *Foxc1^-/-* embryonic (e19.5) cerebellum had additional abnormal granule lineage migratory phenotypes not readily seen in *Foxc1^hith/hith* mice (*Figure 2—figure supplement 2A–H*). In wild-type animals, proliferating GCPs were restricted to the outer EGL (*Figure 2—figure supplement 2A*, inset). However, in *Foxc1^-/-* embryos, we consistently observed streams of proliferating GCPs from the EGL ectopically migrating into the cerebellar core. Some of these streams disrupted the

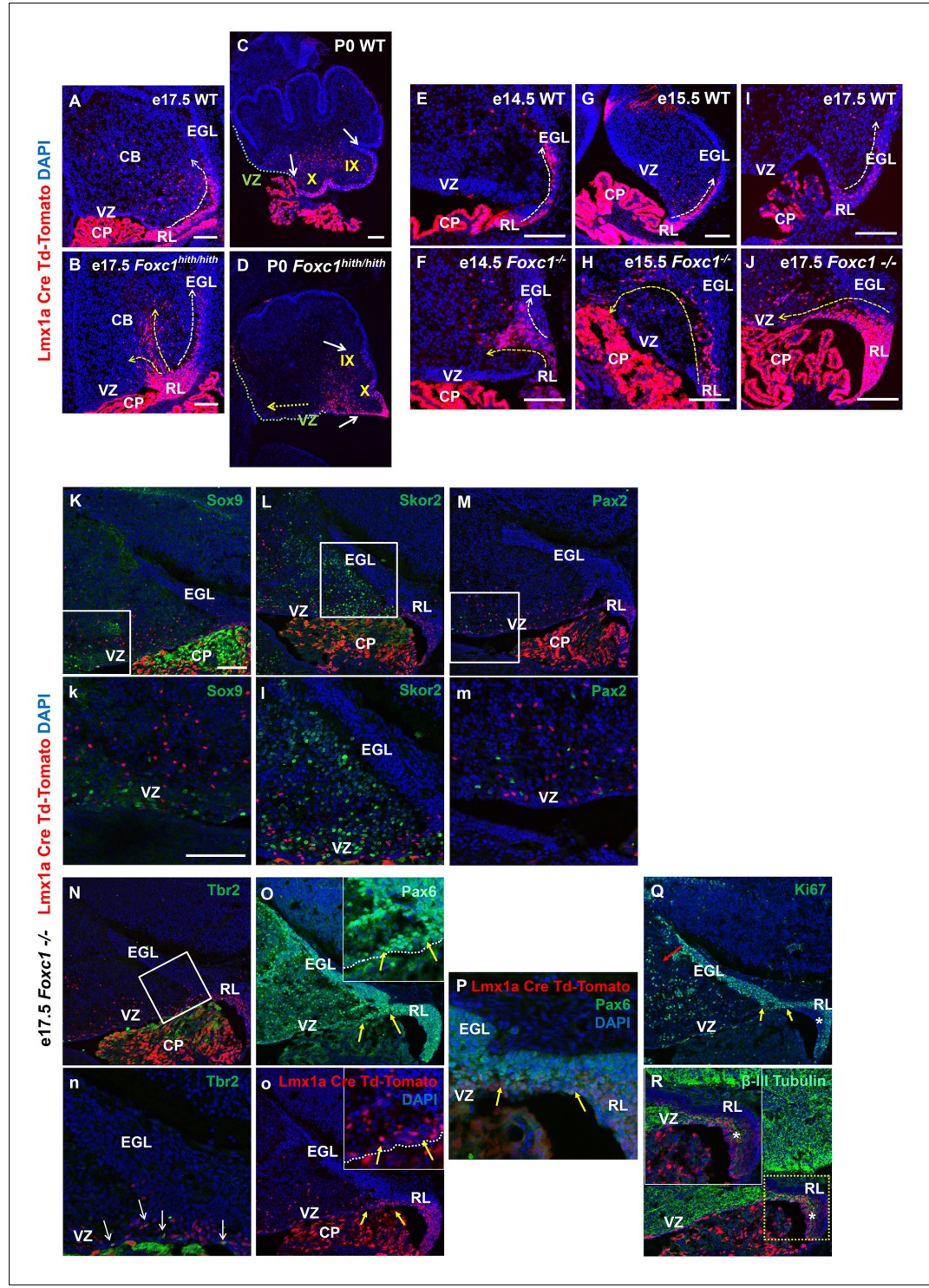

**Figure 2.** Null and Hypomorphic *Foxc1* mutations caused posterior cerebellar foliation defects due to mismigration of cells destined to form the posterior vermis. (**A–J**) Lineage analysis of the *Lmx1a-cre+* cells in the wild-type mice showed tdTomato expression limited to the RL, EGL and presumptive IGL. Postnatally, fate-mapped cells populated the posterior vermis but did not abut the 2° fissure (**C**, white arrows). In the wild-type embryonic cerebellum, these cells were present underneath the EGL directly underneath the pial surface (**A**, **E**, **G**, **I**; white arrow). In *Foxc1^{hith/hith}* *Lmx1a-cre* tdTomato mice, cells migrated out of the RL in multiple ectopic streams (**B**, yellow arrows). Postnatally, in the *Foxc1^{hith/hith}* mutant cerebellum, ectopic tdTomato+ cells were present along the ventricular surface and the inner cerebellar core (**D**, yellow arrows). In *Foxc1^{-/-}* mice (**F,H,J**), *Figure 2 continued on next page*

*Figure 2 continued*

aberrantly migrating *Lmx1a-cre* tdTomato+ cells were evident by e14.5 in the core (**F**) and found in the VZ by e15.5 (**H**, yellow arrow), with an extensive VZ surface presence by e17.5 (**J**; yellow arrow). Additionally, at e17.5, a large number of fate-mapped mutant cells were abnormally retained in an enlarged RL (**J**). None of the mutant internal tdTomato+ cells were Sox9+ (**K,k**), Skor2+ (**L,l**) or Pax2+ (**M,m**), and thus had not undergone a VZ lineage fate-switch. A subset of the fate-mapped cells were Tbr2+ (**N,n**, arrows), as expected of RL-derived unipolar brush cells. All tdTomato+ cells were Pax6+ (**O–P**). This indicated that they retained their RL origin despite aberrant migration. A subset of the Pax6+ cells is Ki67+ (**O, o, Q**; yellow arrows) indicating that they retain their ability to divide, while some tdTomato+ cells in the RL (asterisk) are $\beta$-III Tubulin+ (**R**) and Ki67- indicating that they may have differentiated precociously. *Scale Bar = 100 µm (A–D, K–Q), 50 µm (E–J).*

The following figure supplements are available for figure 2:

**Figure supplement 1.** A significant number of ectopic tdTomato+ cells are found in the *Foxc1* mutant cerebellum.

**Figure supplement 2.** Ectopic populations of granule cell progenitors and Purkinje cells are found in both the *Foxc1*$^{-/-}$ and *Foxc1*$^{hith/hith}$ mutants.

nascent Purkinje cell plate under the EGL, further contributing to the profound disorganization of the *Foxc1*$^{-/-}$ mutant cerebellum (*Figure 2—figure supplement 2B–D*, arrows). Since GCPs secrete reelin as a chemoattractant to Purkinje cells radially migrating outward from their VZ origin, it was not surprising to observe ectopic Purkinje cells within the core of the developing cerebellum together with ectopic granule cells in *Foxc1*$^{-/-}$ mutants (*Figure 2—figure supplement 2F,H*, boxes). We also expect that at least some of these centrally-located ectopic Purkinje cells were the result of aborted outward radial migration of Purkinje cells out of the VZ due to the disruption of the radial scaffold in *Foxc1*$^{-/-}$ mutants, a phenotype that we previously reported (*Haldipur et al., 2014*).

We observed less severe Purkinje cell alignment defects in postnatal *Foxc1*$^{hith/hith}$ cerebella. In wild-type animals, the maturing Purkinje cells organize into a monolayer evident around P16 (*Figure 2—figure supplement 2I*), while in *Foxc1*$^{hith/hith}$ mice, we observed multilayered Purkinje cells in addition to ectopic Purkinje cells fully embedded within the internal granule cell layer (IGL) in multiple lobules (*Figure 2—figure supplement 2J–L*, arrows, box).

Both foliation and lamination abnormalities observed in postnatal *Foxc1*$^{hith/hith}$ mouse mutants are reminiscent of defects observed following meningeal cell destruction by chemical treatment with 6-OHDA in neonatal hamsters (*von Knebel Doeberitz et al., 1986*), which caused disorganized radially oriented glial fibers due to loss of contact with absent meninges. However, in postnatal *Foxc1*$^{hith/hith}$ cerebella, GFAP immunostaining for Bergmann glial fibers showed normal radial fiber organization (*Figure 2—figure supplement 2N*). Laminin immunostaining, which highlights the pial basement membrane, was neither different from wild-type nor discontinuous in early postnatal *Foxc1*$^{hith/hith}$ cerebella (*Figure 2—figure supplement 2P*). Thus, our results do not support a mechanism similar to that of postnatal meningeal ablation for folial and laminar alterations in *Foxc1*$^{hith/hith}$ cerebella. Rather, the *Foxc1*$^{hith/hith}$ phenotype is consistent with a loss of *Foxc1*-dependent SDF1$\alpha$ signaling (*Figure 2—figure supplement 1D*), which we have previously shown to be responsible for a number of embryonic phenotypes, in particular, the loss of radial glial structure and aberrations in cell migration of both the VZ and RL in *Foxc1*$^{-/-}$ mice (*Haldipur et al., 2014*).

## Developmental defects in cerebellar foliation and histogenesis in human fetal del chr 6p25 samples are strikingly similar to those seen in *Foxc1* mutant mice

Although our analysis of cerebellar development in *Foxc1*$^{-/-}$ and *Foxc1*$^{hith/hith}$ mouse mutants has revealed multiple aberrant developmental programs, the relevance of these deficits to human del chr 6p25 DWM is largely inferred. Human brain imaging studies have limited resolution and ideally require autopsy confirmation postmortem. However, few postnatal or fetal human DWM pathology studies are published. Importantly, a recent study by *Delahaye et al. (2012)* identified three human fetal del chr 6p25 DWM cases with large deletions (6.6–17 Mb) encompassing the *FOXC1* gene. The

availability of these rare human fetal samples provided us with the unique opportunity to directly compare cerebellar vermis developmental pathology across species with similar genetic lesions in order to validate our *Foxc1* mutant mouse DWM models.

To study the gross anatomy of the human fetal cerebella, we compared sagittal sections of cerebellar vermis from all three del chr 6p25 fetal cases with age-matched control cases (*Figure 3— source data 1*) (*Delahaye et al., 2012*). No posterior fossa anomalies were detected either by prenatal ultrasound or autopsy in these control cases. Low magnification images of the H and E stained cerebellar vermis from the control cases indicated normal development of the folia. All histogenic layers of the developing cerebellum appeared normal. The EGL was of uniform thickness across the cerebella and the Purkinje cells formed a continuous multilayer Purkinje cell plate under the EGL from the anterior to posterior lobes (*Figure 3A–C*). In all del chr 6p25 cases, striking foliation defects of varying severity were observed (*Figure 3D–F*). In the cerebellum with the least severe DWM phenotype, DW1, the posterior-most lobule, lobule X, was only partially formed (*Figure 3D,G*). However, the anterior lobules appeared relatively normal. The defects in cerebellar foliation were far more severe in the other two del chr 6p25 cases, DW2 and DW3. In both cases, the vermis was hypoplastic with the most severe hypoplasia and dysplasia readily evident in the posterior vermis, despite obvious artefactual tissue damage. Yet, even the lobules in the anterior vermis were not readily distinguishable, with rudimentary fissures defining several lobules of variably indistinct identity. (*Figure 3E,F*). Although the posterior-most vermis was missing in DW3, the remaining posterior surface of DW3 was remarkably flat with no recognizable fissures (*Figure 3F*).

Unique to all of the DW cases, heterotopic cell clusters were evident with H and E staining. These were particularly prominent in DW1 (*Figure 3D,G–I*). For example, at the base of the fissure between lobules IV and V, an ectopic calbindin+ Purkinje cell cluster was evident. Just above this, numerous proliferating Ki67+ Pax6+ GCPs were also apparent. Similar ectopic clusters of Purkinje cells and GCPs were also observed in the other two DW cases (data not shown). A closer analysis of lamination and cellular morphology using calbindin immunohistochemistry to mark Purkinje cells revealed extensive disruptions in all 3 DW cases. In the control cases, the Purkinje cell layer was well-defined, consisting of a distinct multilayer band of cells beneath the molecular layer, above the internal granule layer (*Figure 4A–C*; *Figure 4—figure supplement 1A–C*). In all three del chr 6p25 cases, Purkinje cells were located in a highly disorganized diffuse layer beneath the EGL. This was most marked in the posterior vermis, although in the more severe cases (DW2 and DW3), Purkinje cells were disorganized anteriorly as well (*Figure 4G–I*; *Figure 4—figure supplement 1D–F*). Higher magnification of calbindin-stained sections revealed that in all 3 DW cases, Purkinje cells also lacked their characteristic dendritic arborization, indicating at least a delay in Purkinje cell development. This was consistent with delayed fissure formation in DW2 and DW3. This may indicate a more fundamental abnormality, since Purkinje cell development was abnormal even in DW1 where global developmental delays in foliation were less apparent (Compare *Figure 4—figure supplement 1A–C* with D-F). Ectopic Purkinje cells were also found in the Internal Granule Layer (IGL) and cerebellar white matter (*Figure 3I*).

As expected, all human control samples, regardless of age, had well defined GFAP+ Bergmann glial fibers extending from the EGL to the Purkinje cell layer where their bodies reside (*Figure 4G–I*; *Figure 4—figure supplement 1G–I*). While Bergmann glial fiber morphology was normal in *Foxc1*[*hith/hith*] mouse mutants (*Figure 2—figure supplement 2M,N*), their morphology, number, and location were affected in the *Foxc1*[*-/-*] cerebellum (*Haldipur et al., 2014*). Consistent with *Foxc1*[*-/-*] mouse data, in all three fetal del chr 6p25 cases, fewer Bergmann glial fibers were evident, with some regions devoid of Bergmann glia fibers entirely (*Figure 4J–L*; *Figure 4—figure supplement 1J–L*). Remaining Bergmann glial fibers were severely dysmorphic. This may be related to the direct requirement of SDF1α to maintain cerebellar Bergmann glial fibers (*Haldipur et al., 2014*). Purkinje cell deficits may also contribute to this phenotype, since radial glial maintenance also depends on normal Purkinje cell development (*Dahmane and Ruiz i Altaba, 1999*).

The striking similarities in phenotype between *Foxc1* mouse mutant cerebella and the human del chr 6p25 DWM cases validate our mouse *Foxc1* mutant mouse models and suggest that many of the key mechanisms controlling cerebellar development are conserved between mouse and human.

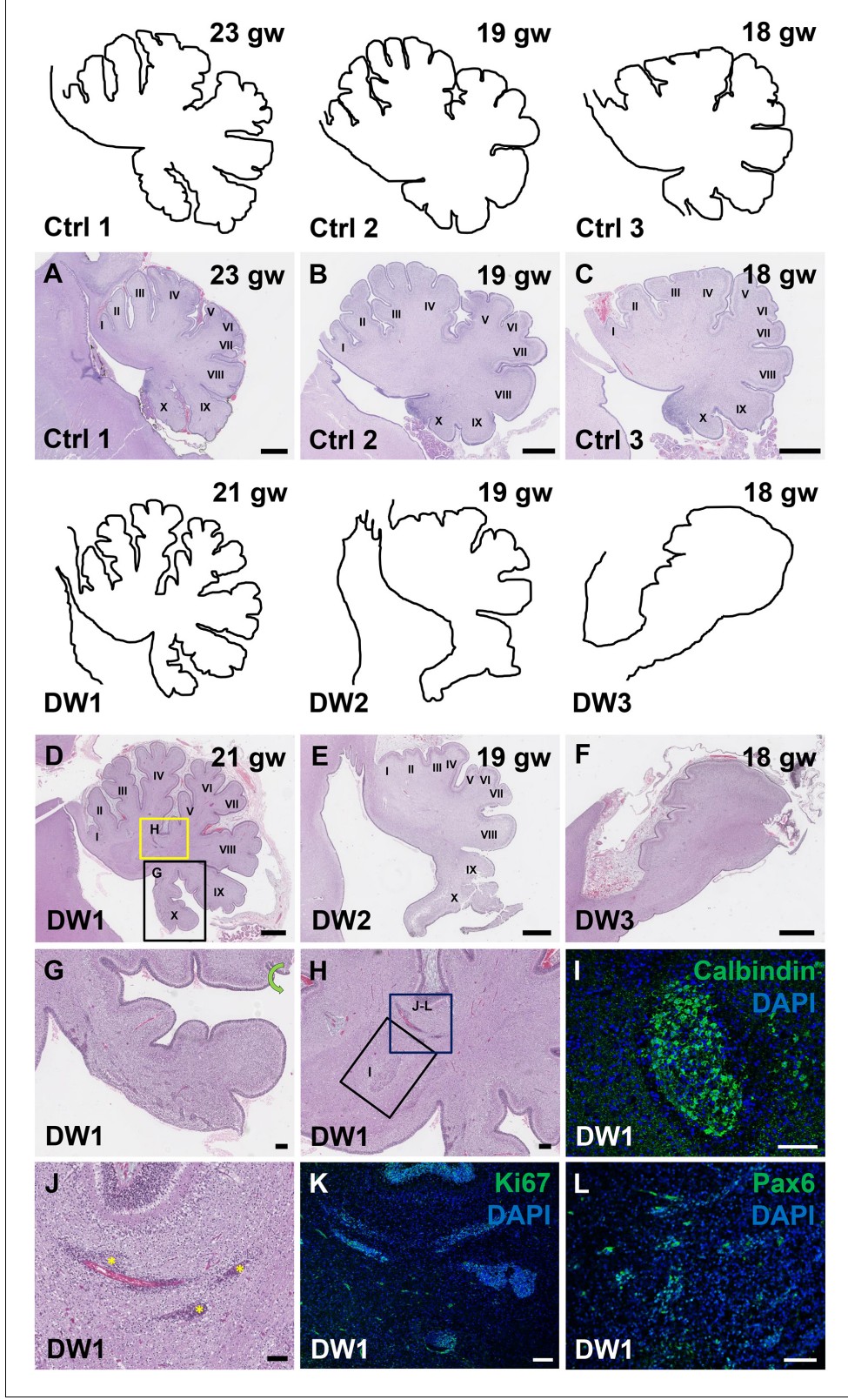

**Figure 3.** Defects in cerebellar foliation and histogenesis were observed in human DWM cases with deletions in chromosome 6p25. (A–I) Hematoxylin and Eosin stained midsagittal sections through the fetal cerebellar vermis in normal (A–C) and del chr 6p25 samples (D–F). Ages are indicated in gestational weeks (gw). Cerebellar outlines are provided for clarity with higher magnification locations indicated. The X[th] lobule of the posterior vermis in
*Figure 3 continued on next page*

*Figure 3 continued*

DW1 (**D**, black box; **G**) was only partially formed, similar to the *Foxc1^hith/hith^* cerebellum while in (**E–F**), the posterior vermis was severely dysplastic in DW2 and DW3. The del chr 6p25 cerebella also had ectopic Calbindin+ Purkinje cells (**H**, box and **I**) and ectopic Ki67+ Pax6+ GCPs (**H**, box **J–L**). *Scale bar = 1 mm (**A–F**), 200 µm (**G**) and 100 µm (**H–L**).*

The following source data is available for figure 3:

**Source data 1.** List of control and Dandy-Walker malformation cases listed in the study.

## Discussion

Although the diagnosis of DWM has improved with advances in brain imaging (*Doherty et al., 2013*), the mechanisms leading to this important human cerebellar malformation remain largely unknown. A few genes have been implicated in rare cases of DWM (*Aldinger and Doherty, 2016*). Of these, *FOXC1* is the best studied (*Aldinger et al., 2009*; *Haldipur et al., 2014*), with mutant mouse data demonstrating several important roles for *Foxc1* in early mouse cerebellar anlage development. Notably, *Foxc1* expression is largely limited to the posterior fossa mesenchyme rather than the developing cerebellum itself. Complete loss of *Foxc1* in mice (homozygous null animals) leads to loss of *Foxc1*-dependent expression of SDF1α, a secreted factor in the posterior mesenchyme. We

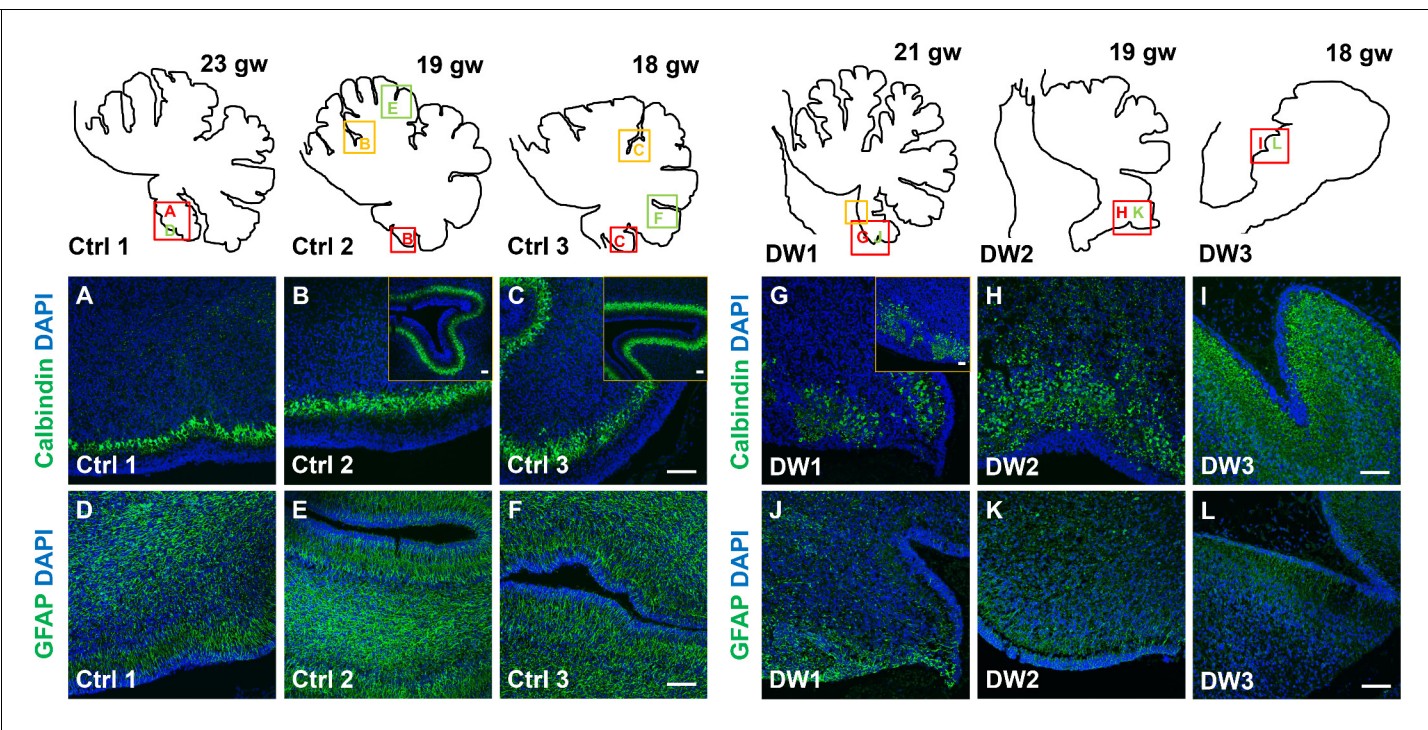

**Figure 4.** Defects in Purkinje cell alignment and Bergmann glial morphology were observed in the cerebellum of fetuses with chr 6p25 mutations. (**A–L**) Midsagittal sections through normal (**A–F**) and del chr 6p25 (**G–L**) fetal cerebellar vermis stained for Calbindin (**A–C**, **G–I**) and GFAP (**D–F**, **J–L**). Cerebellar outlines are provided for clarity. Colored boxes show locations of panels. Ages are indicated in gestational weeks (gw). Purkinje cells during normal cerebellar development were arranged in a distinct multilayered band beneath the molecular layer throughout the cerebellum from 18–23 gw (**A–C**, insets). In all del chr 6p25 cases, Purkinje cells were ectopically broadly distributed in the forming cerebellar cortex (**G–I**). Bergmann glial fibers extended from the PC layer to the EGL in the normal cerebellum (**D–F**). These fibers were sparse and highly dysmorphic in the del chr 6p25 cases (**J–L**). *Scale bar = 100 µm.*

The following figure supplement is available for figure 4:

**Figure supplement 1.** Purkinje cell and Bergmann glial fiber morphologies were disrupted in human fetal del chr 6p25 cases.

previously showed that SDF1α acts via its receptor, Cxcr4, which is expressed in cerebellar radial glial cells and is required for both proliferation of these VZ progenitors in addition to maintenance of their radial fibers, which are the migration scaffold for cerebellar GABAergic neurons to reach the cerebellar cortex (*Haldipur et al., 2014*). Notably, the *Foxc1*[-/-] phenotype is much more severe than that seen in human DWM patients (del chr 6p25), who harbor a deletion of the *FOXC1* gene, yet retain one intact copy on an unaffected chromosome (*Aldinger et al., 2009*; *Delahaye et al., 2012*). *Foxc1*[+/-] mice have no cerebellar phenotype, however, *Foxc1*[hith/hith] are postnatal viable and have disrupted cerebellar morphology (*Aldinger et al., 2009*). We have now shown that *Foxc1*[hith/hith] mutants display a unique posterior vermis foliation defect that is strikingly similar to that shared with all human del chr 6p25 DWM patients. Using lineage fate mapping in mouse models, we demonstrated that embryonic migration abnormalities of the posterior-fated cerebellar RL descendants causes this phenotype in both *Foxc1*[hith/hith] and *Foxc1*[-/-] mice and showed that *Foxc1*[hith/hith] mice also display defects in postnatal cerebellar lamination. Finally, we directly compared the developing mouse *Foxc1* mutant phenotypes with very rare human del chr 6p25 fetal cases, the first reported in-depth analysis of fetal DWM samples. Strikingly, the human fetal DWM cases and our mouse models share very similar developmental pathogenesis, validating our mouse models and demonstrating that many of the key developmental mechanisms are conserved between the two species.

Although cerebellar vermis dysplasia is highly variable in MRI of postnatal human del chr 6p25 DWM (*Delahaye et al., 2012* and *Figure 1*), all three fetal cases presented here have a prominent elongation of a dysplastic posterior vermis 'tail'. This 'tail sign' has recently been proposed to be a pathognomonic feature for all DWM (*Bernardo et al., 2015*), irrespective of those with a known genetic cause, such as del chr 6p25. Strikingly, the human DWM tail is highly reminiscent of the partially formed, unpaired posterior lobule we observed in the postnatal *Foxc1*[hith/hith] mutant mouse cerebellar vermis. This phenotype appears to be unique as we are unaware of any other mouse mutant with this foliation defect. Through lineage tracing experiments, we showed extensive ectopic migration of posterior-fated cerebellar RL descendants in both *Foxc1*[hith/hith] and *Foxc1*[-/-] mice. These ectopic paths included the mutant VZ. Cerebellar progenitor fate switches have been implicated in human cerebellar agenesis (*Pascual et al., 2007*; *Millen et al., 2014*). We therefore tested if the mutant ectopic RL derived cells in the VZ or anlage showed evidence of a RL to VZ fate switch. However, all *Lmx1a-cre* lineage labeled cells maintained Pax6 expression indicative of a RL lineage identity. We conclude that the *Foxc1* mutant posterior vermis phenotype is mostly due to misguided migration of RL-derived cells.

SDF1α expressed in the head mesenchyme is a direct transcriptional target of Foxc1 (*Zarbalis et al., 2012*). We have demonstrated that loss of SDF1α can rescue *Foxc1*[-/-] cerebellar phenotypes (*Haldipur et al., 2014*). At e12.5, when Foxc1 expression is initiated in the posterior fossa mesenchyme overlying the cerebellum, SDF1α expression is significantly downregulated in both *Foxc1*[-/-] and *Foxc1*[hith/hith] embryos at e12.5 (*Aldinger et al., 2009*) (*Figure 2—figure supplement 1D*). Since SDF1α is expressed in the mouse posterior fossa mesenchyme prior to e12.5 ([*Lein et al., 2007*], Website: © 2015 Allen Institute for Brain Science. Allen Mouse Brain Atlas [Internet]. Available from: http://mouse.brain-map.org.), we conclude that *Foxc1* is required to maintain, but not initiate, SDF1α. This maintenance role is key to understanding the increased vulnerability of the posterior vermis in *Foxc1* mouse mutants and human del chr 6p25 DWM. Previous studies have shown that SDF1α secreted by embryonic meningeal cells embryonically acts as a chemoattractant, regulating the tangential migration of Cxcr4+ GCPs away from the RL to form the EGL (*Yu et al., 2010*; *Haldipur et al., 2014*). The cerebellar RL generates assorted glutamatergic brain stem nuclei and cerebellar nuclei in the mouse at e10.5 and anteriorly fated GCPs just before e12.5 (*Machold and Fishell, 2005*). By e12.5, only posterior vermis-fated GCPs and unipolar brush cells remain in the RL to emerge over the remaining days of mouse embryogenesis. Our new data clearly demonstrates that these late derivatives are absolutely dependent on continued SDF1α expression which itself is dependent on *Foxc1*. Our data also confirm previous studies showing a role for SDF1α as a chemoattractant in the developing pia to maintain GCPs within the proliferating outer zone of the developing EGL adjacent to the pial surface (*Hartmann et al., 1998*; *Wiegand et al., 1998*; *Zou et al., 1998*; *Reiss et al., 2002*; *Zhu et al., 2002*; *Vilz et al., 2005*; *Tiveron and Cremer, 2008*; *Yu et al., 2010*). In *Foxc1* mutants, SDF1α is not maintained at appropriate levels, leading to an additional phenotype of ectopic proliferating EGL cells within the cerebellar anlage in both the anterior and posterior vermis. Our results add to the growing body of evidence that the posterior fossa

mesenchyme, through SDF1α expression, preserves the structure of both the RL and the nascent EGL. By ensuring that cells correctly exit the RL and remain within the GCP niche of the EGL, the mesenchyme regulates the proper formation and lamination of cerebellar folia (*Figure 5*).

To our knowledge, this report represents the first in-depth analysis of human fetal DWM. Additionally, since the cases of human del chr 6p25 we analyzed are specifically modeled by our *Foxc1* mutant mice, a direct phenotypic comparison between the two species with correlated genotypes was possible. Direct comparisons between developing human fetal tissue with known genetic lesions and mouse developmental models of comparable genotypes are extremely rare in literature. All three human del chr 6p25 fetal DWM fetal cases we assessed here exhibited a striking dysplasia of the posterior vermis and also had ectopically located GCPs and Purkinje cells and dysmorphic and sparse Bergmann glial fibers. The phenotypes observed in the del chr 6p25 cerebella are similar to those observed in the *Foxc1*⁻/⁻ mouse, providing ample evidence of the validity of our *Foxc1*⁻/⁻ DWM models.

It is necessary to state the difficulties and drawbacks of this study whilst highlighting the legitimacy and need for such analyses. First, it is clear that some of the mechanisms we have attributed to posterior vermis hypoplasia in human DWM based on our *Foxc1* mouse models have not been directly validated in the human cerebellum. The human fetal DWM cases presented here are snapshots of three separate dysmorphic cerebella at specific stages of development (18–21 gw). Given the clinical constraints of human fetal tissue research and the rarity of genetically defined malformation cases, a thorough study of multiple stages of abnormal prenatal cerebellar development will be nearly impossible. We cannot define the developmental progression of these specific cases with their respective phenotypes, nor can we predict their phenotypic outcomes if the pregnancies had continued. Further, since DWM is not currently readily diagnosed by imaging until 18 gw, it is likely impossible to obtain earlier cases. However, human fetal analysis, as we have presented here, is essential because it is the only analysis that is possible. Animal models are required to investigate potential mechanisms and our current work clearly validates our *Foxc1* mutant mouse models. Since our mouse models indicate that human del chr 6p25 DWM is largely due to a disruption of early mesenchymal signaling and aberrant rhombic lip development, a long term goal is to develop human fetal

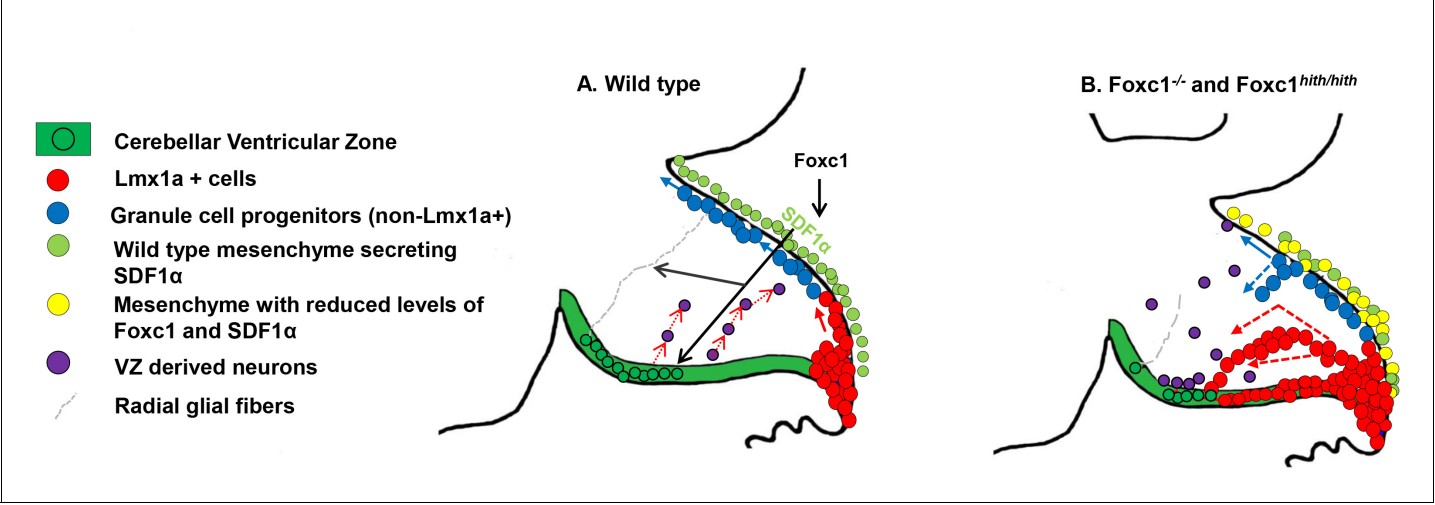

**Figure 5.** Summary of Foxc1-dependent molecular and cellular mechanisms contributing to del chr 6p25 cerebellar phenotypes. (**A**) Schematic of a paramedial sagittal section of the embryonic mouse cerebellum. In the wild-type cerebellum, mesenchymal Foxc1 controls the expression of chemokine SDF1α which binds to its receptor Cxcr4 which is strongly expressed in the RL, EGL, and VZ. SDF1α functions as a chemoattractant to Lmx1a+ (red) and Lmx1a- (blue) GCPs exiting the RL to form the EGL, ensuring that these progenitors exit the RL and remain confined to the EGL underneath the pial surface. SDF1α also controls the migration of cells out of the VZ, acting as a chemoattractant. It is also required for the maintenance of radial glial fibers, which act as scaffolds for this migration. (**B**) In the *Foxc1*⁻/⁻ and *Foxc1*^hith/hith^ mice, deletion of *Foxc1* leads to a significant downregulation of mesenchymal SDF1α by e12.5. This reduction results in excessive retention of posterior-fated cells in the RL and ectopic migration of cells out of the RL (red arrows) and precocious migration of GCPs from the EGL into the cerebellar anlage (blue arrows). Proliferation, migration, and VZ-derived neurons and radial glia are also negatively affected.

imaging technology and protocols targeted to these phenotypes to facilitate earlier detection and perhaps provide therapeutic intervention.

In conclusion, this study presents an analysis of the changes that take place during mouse and human cerebellar development following the loss of *Foxc1* and subsequent disruptions in mesenchymal signaling. We show that early disruption in mesenchymal signaling has immediate effects, including mismigration of Purkinje cells and GCPs from both the RL and EGL. There are both direct and indirect repercussions on later developmental events that lead to abnormal foliation, posterior vermis hypoplasia, developmental delays, and abnormal layering of Purkinje cells. Together our human and mouse analyses provide compelling evidence to support our model of Foxc1 control of cerebellar development and developmental pathogenesis of *FOXC1*-dependent DWM (*Figure 5*). Notably, however, human del chr 6p25 DWM cases are rare. An essential question remains as to whether the key features and mechanisms we have elucidated for *Foxc1*-dependent DWM are generalizable to all forms of DWM, regardless of the underlying genotype.

# Materials and methods

## Subjects

All human studies were approved by Institutional Review Boards at all participating institutions. Written informed consent was obtained from all subjects. Criteria for diagnosing DWM included a) cerebellar vermis hypoplasia affecting the posterior vermis more severely than the anterior, b) an enlarged fourth ventricle, c) the upward rotation of the cerebellar vermis and an enlarged posterior fossa. Genetic analysis for all DWM cases has previously been published (*Aldinger et al., 2009*; *Delahaye et al., 2012*).

## Animals

*Lmx1a-cre* (*Chizhikov et al., 2010*) and *Ai14* (*B6.Cg-Gt (ROSA) 26Sortm14 (CAG-tdTomato)Hze/J*; RRID:IMSR_JAX:007914 Jackson Laboratories Stock Number: 007914, N5 +F13 as of 05.05.2015) (*Madisen et al., 2010*) were used for fate mapping studies. All mouse tissue was processed as previously published (*Haldipur et al., 2014*).

## Human fetal tissue

All human fetal samples (*Figure 3—source data 1*) were obtained in accordance with approved IRB protocols from SCRI. Control samples were obtained from the Department of Laboratories, Seattle Children's Hospital, Seattle, USA and Department of Pathology, Hospital S. Giovanni di Dio, e Ruggi d'Aragona, Salerno, Italy. These were from the fetuses of elective or spontaneous terminations or intrauterine fetal deaths. Only tissues with no detectable cerebellar pathology following histopathological analyses were included as controls. The three fetal human del chr 6p25 DWM cerebellar samples were obtained from Hôpital Robert-Debré, Paris, France (*Delahaye et al., 2012*).

## Histology and immunohistochemistry

The primary antibodies used in this study were Calbindin (Swant, CB38, Switzerland; 1:3000; RRID: AB_10000340), Foxp2 (Everest Biotech, EB05226 – 1:1000; RRID:AB_2107112), Ki67 (Vector – 1:300; RRID:AB_2336545), Laminin (Sigma, L9393 – 1:25; RRID:AB_477163), GFAP (DAKO, Z0034 – 1:1000; RRID: AB_10013482), Pax6 (Covance, 901301–1:200; RRID: AB_2315069), Pax2 (Zymed, 7160000–1:200; RRID:AB_2533990), Skor2 (Novus Biologicals, NBP2-14565 1:100; RRID:AB_2632379), Lmx1a (Millipore AB10533, 1:2000; RRID:AB_10805970) and Tbr2 (gifted by Robert Hevner; 1:1000; RRID: AB_2315446). We were unable to carry out double IHCs with tdTomato and Pax6/Ki67 because both primary antibodies function only following antigen retrieval which bleaches out the tdTomato fluorescence. Hence, we resorted to imaging the section with tdTomato marked cells prior to antigen retrieval and the same section was imaged following incubation with Pax6 or Ki67. For Pax6, we have presented an overlay of tdTomato and Pax6 IHC images of the same section, although they were not captured simultaneously (*Figure 2O–P*).

## Cell counts and data analyses

To evaluate Lmx1a expression in the RL, the total number of Lmx1a positive cells in the RL was counted. This was followed by a total DAPI count that represented the total cell count in the RL. The percentage of DAPI+ cells in the RL that were also Lmx1a+ was represented in the graph. We also quantified the number of tdTomato+ cells that are found outside the RL and EGL area from e14.5-e17.5 in midsagittal sections. Statistical tests were the same as previously applied.

## RNA extraction and quantitative real-time RT-PCR

RNA was extracted from E12.5 hindbrain of wild-type (n = 4) and Foxc1$^{hith/hith}$ (n = 4) littermate embryos. qRT-PCR was performed with 4–6 biological replicates as described (*Aldinger et al., 2009*).

## Acknowledgement

We thank Dr. Tom Kume (Northwestern University, Chicago, IL) and Samuel J Pleasure (UCSF) for providing us with the *Foxc1$^{-/-}$* and *Foxc1$^{hith/hith}$* mouse strains respectively. We also gratefully acknowledge the technical assistance of Joanna Yeung, Arianna Gomez, Conrad Winter, and Gwendolyn S Gillies. We thank Dr. Alexandra J Joyner (Sloan-Kettering Memorial Hospital, New York) and Dr. Raj Kapur (Seattle Children's Hospital) for valuable discussions. The work described herein was supported by National Institutes of Health R01NS072441 and R01NS080390 to KJM. All co-authors have seen and agreed to the contents of this manuscript. None of the co-authors have any potential financial interests or conflict of interest with respect to this manuscript.

## Additional information

### Funding

| Funder | Grant reference number | Author |
| --- | --- | --- |
| National Institutes of Health | R01NS072441 | Kathleen J Millen |
| National Institutes of Health | R01NS080390 | Kathleen J Millen |

The funders had no role in study design, data collection and interpretation, or the decision to submit the work for publication.

### Author contributions

PH, Conceptualization, Data curation, Formal analysis, Investigation, Visualization, Methodology, Writing—original draft, Writing—review and editing; DD, Data curation, Formal analysis, Validation, Methodology, Writing—review and editing; KAA, Data curation, Formal analysis, Validation, Investigation, Methodology; OKJ, Data curation, Methodology; FG, HA-B, Resources, Writing—review and editing, FG and HAB provided us with cerebellar samples from fetuses with 6p25 del; WBD, Resources, Formal analysis, Methodology, Writing—review and editing; JRS, RR, Resources, Writing—review and editing, provided us with cerebellar samples from normal fetuses with out any cerebellar abnormalities; KJM, Conceptualization, Formal analysis, Supervision, Funding acquisition, Investigation, Visualization, Methodology, Writing—original draft, Writing—review and editing

### Author ORCIDs

Kathleen J Millen, http://orcid.org/0000-0001-9978-675X

### Ethics

Human subjects: All human studies were approved by Institutional Review Boards at all participating institutions. Written informed consent was obtained from all subjects.

Animal experimentation: All animal experimentation for this study was approved by the Institutional Animal Care and Use Committee (IACUC Protocol no 14208), of Seattle Children's Research Institute, Seattle, WA, USA.

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
