## [Decision Letter]

Thank you for submitting your article "Mouse and Human *Foxc1* dependent Dandy-Walker cerebellar malformation share common mechanisms" for consideration by *eLife*. Your article has been reviewed by two peer reviewers, and the evaluation has been overseen by Robb Krumlauf as a Reviewing Editor and a Senior Editor. The reviewers have opted to remain anonymous.

The reviewers have discussed the reviews with one another and the Reviewing Editor has drafted this decision to help you prepare a revised submission.

Summary and essential revisions:

This manuscript details phenotypic overlaps in the developing cerebellum of *Foxc1* mutant mice and human fetus with del chr 6p25 supporting the idea that *Foxc1* mutant mice are an excellent model for Dandy-Walker malformation (DWM). The authors have identified a previously uncharacterized migration defect of granule cell progenitors in *Foxc1* mutant mice suggesting that *Foxc1* regulates expansion and foliation of the posterior lobe via signals originated from the posterior fossa mesenchyme in mice and humans. This study is novel by providing in-depth and direct comparisons between the human fetal brain and mouse models for disease DWM and is also valuable for insight on the degree of conservation in mechanisms controlling cerebellar development between humans and mice. Therefore, it is worthy of consideration for publication in *eLife*. However, there are a number of specific concerns shared by the reviewers primarily related problems in the depth of the characterization of mouse *Foxc1* mutant cerebella that must be addressed before publication can be considered.

Specific comments:

1) The authors show that many tdTomato-labeled descendants of the Lmx1a lineage are abnormally present in the cerebellar ventricular zone (VZ). The data provided are insufficient to determine if *Foxc1* mutation causes a cell-fate change in the cerebellar VZ. It is also unclear whether the mutation affects the expression of Lmx1a or abnormal migration of the Lmx1 lineage. The authors claim that Pax6 is present in most tdTomato+ cells, and argue against abnormal cell fate specification. However, the quality of Figure 2 is too poor to support this conclusion. The authors need to provide improved Pax6/tdTomato double labeling and examine additional markers, for example, Zic1 for granule cells and Sox9 or Sox2 for VZ radial glia. In addition, short-term fate mapping or examination of Lmx1a expression should be provided to show whether *Foxc1* mutation affects Lmx1a expression.

2) It is unclear how abnormal migration of the Lmx1a-lineage causes the unusual foliation defect in the *Foxc1* mutant cerebellar vermis. Do the misrouted granule cell progenitors fail to proliferate? Is the VZ in the posterior cerebellar anlage compromised in the mutants?

3) In Figure 2, a conspicuous phenotype of *Foxc1* mutant cerebella seems to be the increased contribution of tdTomoto+ cells to the choroid plexus (which appears larger than the control). Is this phenotype real? As the rhombic lip (also known as the germinal trillion) links to the VZ, external granule layer, and choroid plexus, the authors may examine if the rhombic lip increases contribution to the choroid plexus at the expense of the VZ and EGL in *Foxc1* mutant cerebella.

4) Why are only the late RL derivatives affected? I am not entirely sure an explanation is given for this, and this is critically important because, apparently, the same population is affected in mouse and human. Is this a timing issue, or a gene expression/function issue? Either way, it appears that a highly conserved feature of cerebellum development is affected in both species.

5) This study primarily focuses on the posterior lobe (lobule X) defects and needs to be expanded to discuss other defects. The mouse Figure 1 (panel I) and human Figure 3 (panels D and E) indicates the anterior lobule is also severely affected. Based on the limited histological data on the DWM cerebella (including the previously published ones), the claim that the partially formed posterior lobe in the *Foxc1* mutant cerebella echoes the posterior vermis DW "tail sign" seems to be an overstatement. The Bernardo et al. paper (2015) does not explicitly claim a link between the "tail sign" and defects specific to the posterior lobe of DW cerebellar vermis. The authors do not make a clear connection between the "tail sign" defined by MRI imaging and the histological defect specific to the posterior lobe. However, they state in the Abstract that "Particularly striking is the presence of a partially formed posterior lobe echoing the posterior vermis DW 'tail sign' observed in human imaging studies." Other defects, such as abnormal arrangement of Purkinje cells and Bergmann glial fibers and abnormal folding of the cerebellar cortex, are quite generic phenotypes.

6) Based on Figure 2 it looks to me that there may be more Cre marked cells in the mutant posterior cerebellum compared to control. Is this true? And, it also looks like equal numbers of cells adjacent to the "green" VZ line in mutant and control. Apologies if I have missed it, but is there a quantification for these data? It is hard to appreciate that there are ectopic RL derived cells near the VZ.

---

## [Author Response]

*Specific comments:*

*1) The authors show that many tdTomato-labeled descendants of the Lmx1a lineage are abnormally present in the cerebellar ventricular zone (VZ). The data provided are insufficient to determine if Foxc1 mutation causes a cell-fate change in the cerebellar VZ. It is also unclear whether the mutation affects the expression of Lmx1a or abnormal migration of the Lmx1 lineage. The authors claim that Pax6 is present in most tdTomato+ cells, and argue against abnormal cell fate specification. However, the quality of Figure 2 is too poor to support this conclusion. The authors need to provide improved Pax6/tdTomato double labeling and examine additional markers, for example, Zic1 for granule cells and Sox9 or Sox2 for VZ radial glia. In addition, short-term fate mapping or examination of Lmx1a expression should be provided to show whether Foxc1 mutation affects Lmx1a expression.*

We determined that the expression of Lmx1a in the RL at e13.5, shortly after its initiation, is unchanged in *Foxc1^-/-^*cerebellum compared to the wildtype controls (Figure 2—figure supplement 1). This strengthens our conclusion that the phenotype we observe in the *Foxc1* mutants is a result of abnormal distribution of the *Lmx1a* RL lineage.

We replaced the panel in Figure 2 with better images at earlier stages to show a stream of abnormally migrating cells. The new images (Figure 2) clearly indicate that none of the Lmx1a-Cre td Tomato+ cells migrating away from the RL stain for markers of VZ progenitors (Sox9) or VZ derivatives (Pax2, Skor2). The mismigrating tdTomato+ cells continue to express Pax6 indicating that they retain their RL and granule cell identity (Figure 2). A subset of these Pax6 positive cells continues to divide but some begin to differentiate (Figure 2).

*2) It is unclear how abnormal migration of the Lmx1a-lineage causes the unusual foliation defect in the Foxc1 mutant cerebellar vermis. Do the misrouted granule cell progenitors fail to proliferate? Is the VZ in the posterior cerebellar anlage compromised in the mutants?*

The direct link between abnormal posterior foliation and RL abnormalities is not entirely clear, however, as we showed in Chizhikov et al., 2010, PNAS, PMID#20498066, the posterior vermis is a derivative of the late RL. The late RL is compromised in both *Foxc1^-/-^*and *FoxC1^hith/hith^*mutants.

A large subset of the misrouted GCPs continues to proliferate (Figure 2; Figure 2—figure supplement 2). However, we observe that many of the *Lmx1aCre* td-Tomato+ cells that fail to migrate out of the RL eventually differentiate precociously within the RL (Figure 2). A combination of mismigration and precocious differentiation depletes the progenitors that are required to form the posterior vermis, thus leading to the formation of severely hypoplastic posterior lobules.

We have now included data demonstrating that the VZ is not compromised in *FoxC1^hith/hith^*(Figure 2—figure supplement 1). However, we have previously published that the VZ is affected in the null mutants (Haldipur et al., 2014, eLife, PMID#25513817).

*3) In Figure 2, a conspicuous phenotype of Foxc1 mutant cerebella seems to be the increased contribution of tdTomoto+ cells to the choroid plexus (which appears larger than the control). Is this phenotype real? As the rhombic lip (also known as the germinal trillion) links to the VZ, external granule layer, and choroid plexus, the authors may examine if the rhombic lip increases contribution to the choroid plexus at the expense of the VZ and EGL in Foxc1 mutant cerebella.*

The phenotype described by the reviewer is indeed real. We do find a much larger choroid plexus in the mutants (Aldinger et al., 2009, Nature Genetics, PMID#19668217). We are unable to directly test this hypothesis due to the lack of RL-specific Cre mouse models or gene enhancer elements with RL expression not also accompanied by choroid plexus expression.

*4) Why are only the late RL derivatives affected? I am not entirely sure an explanation is given for this, and this is critically important because, apparently, the same population is affected in mouse and human. Is this a timing issue, or a gene expression/function issue? Either way, it appears that a highly conserved feature of cerebellum development is affected in both species.*

In the Discussion, we described our hypothesis for why we think late RL derivatives are more susceptible to decreased *Foxc1* expression (third paragraph). Anterior-fated lineages migrate out of the RL prior to e13.5 and are hence less susceptible to the effects of *Foxc1* deletion. We have clarified this in the text.

*5) This study primarily focuses on the posterior lobe (lobule X) defects and needs to be expanded to discuss other defects. The mouse Figure 1 (panel I) and human Figure 3 (panels D and E) indicates the anterior lobule is also severely affected. Based on the limited histological data on the DWM cerebella (including the previously published ones), the claim that the partially formed posterior lobe in the Foxc1 mutant cerebella echoes the posterior vermis DW "tail sign" seems to be an overstatement. The Bernardo et al. paper (2015) does not explicitly claim a link between the "tail sign" and defects specific to the posterior lobe of DW cerebellar vermis. The authors do not make a clear connection between the "tail sign" defined by MRI imaging and the histological defect specific to the posterior lobe. However, they state in the Abstract that "Particularly striking is the presence of a partially formed posterior lobe echoing the posterior vermis DW 'tail sign' observed in human imaging studies." Other defects, such as abnormal arrangement of Purkinje cells and Bergmann glial fibers and abnormal folding of the cerebellar cortex, are quite generic phenotypes.*

Indeed, we observe anterior defects. However, the anterior lobe is relatively normal in terms of histogenesis compared to the posterior vermis, which is far more hypoplastic and dysplastic. We chose to focus on the posterior vermis since, to our knowledge, the posterior *FoxC1^hith/hith^*foliation is unique and unusual, and the hypoplastic posterior vermis with a partially formed unpaired posterior lobule seems to be a common link between our mouse model, the human samples presented in our study, as well as the limited number of Dandy-Walker pathology based studies (Bernardo et al., 2015, Prenat Diagn, PMID#26448595; Russo and Fallet-Bianco, 2007, J Child Neurol, PMID#17621537, Kapur et al., 2009, Birth Defects Res A Clin Mol Teratol, PMID# 19441098). The reviewer is correct in pointing out that the tail sign described by Bernardo et al. is not linked to posterior vermis hypoplasia in their study. However, based on the limited H&E images presented in Bernardo et al., it is clear that the tail sign they are referring to is the posterior lobule, which is clearly unpaired and partially formed. This is very much like what we observe in *FoxC1^hith/hith^*mutants and is readily seen in the human DW1 control presented here. Our mouse functional studies also show that this specific disruption of patterning and foliation in the posterior vermis is due to RL abnormalities.

*6) Based on Figure 2 it looks to me that there may be more Cre marked cells in the mutant posterior cerebellum compared to control. Is this true? And, it also looks like equal numbers of cells adjacent to the "green" VZ line in mutant and control. Apologies if I have missed it, but is there a quantification for these data? It is hard to appreciate that there are ectopic RL derived cells near the VZ.*

We have replaced the image in Figure 2 with a better image that is a more accurate indicator of the distribution of Cre+ cells in the mutant. We do think that there are more tdTomato+ cells along the VZ of the mutant cerebellum when compared to WT, although to better appreciate the extent of mismigration of RL derivatives in the mutant, we now quantified the number of tdTomato+ cells that are found outside the RL and EGL area from e14.5-e17.5 in midsagittal sections. The numbers in the WT are representative of the unipolar brush cell population as well as some GNs that have differentiated and migrated into the IGL. However, we do see a striking increase in the number of tdTomato+ cells outside the RL and EGL in *Foxc1* mutants; many of which are ectopic in nature (Figure 2; Figure 2—figure supplement 1).